# Novel Bacteriophage Specific against *Staphylococcus epidermidis* and with Antibiofilm Activity

**DOI:** 10.3390/v14061340

**Published:** 2022-06-20

**Authors:** Rima Fanaei Pirlar, Jeroen Wagemans, Luis Ponce Benavente, Rob Lavigne, Andrej Trampuz, Mercedes Gonzalez Moreno

**Affiliations:** 1Charité—Universitätsmedizin Berlin, Corporate Member of Freie Universität Berlin, Humboldt-Universität zu Berlin, Center for Musculoskeletal Surgery, Augustenburger Platz 1, 13353 Berlin, Germany; rima.fanaei@charite.de (R.F.P.); luis.ponce-benavente@charite.de (L.P.B.); andrej.trampuz@charite.de (A.T.); 2Berlin Institute of Health at Charité, Universitätsmedizin Berlin, BIH Center for Regenerative Therapies (BCRT), Charitéplatz 1, 10117 Berlin, Germany; 3KU Leuven, Department of Biosystems, Kasteelpark Arenberg 21, 3001 Leuven, Belgium; jeroen.wagemans@kuleuven.be (J.W.); rob.lavigne@kuleuven.be (R.L.)

**Keywords:** *Staphylococcus epidermidis*, biofilm-associated infection, novel bacteriophage species, whole-genome sequencing, isothermal microcalorimetry

## Abstract

*Staphylococcus epidermidis* has emerged as the most important pathogen in infections related to indwelling medical devices, and although these infections are not life-threatening, their frequency and the fact that they are extremely difficult to treat represent a serious burden on the public health system. Treatment is complicated by specific antibiotic resistance genes and the formation of biofilms. Hence, novel therapeutic strategies are needed to fight these infections. A novel bacteriophage CUB-EPI_14 specific to the bacterial species *S. epidermidis* was isolated from sewage and characterized genomically and phenotypically. Its genome contains a total of 46,098 bp and 63 predicted genes, among which some have been associated with packaging and lysis-associated proteins, structural proteins, or DNA- and metabolism-associated proteins. No lysogeny-associated proteins or known virulence proteins were identified in the phage genome. CUB-EPI_14 showed stability over a wide range of temperatures (from −20 °C to 50 °C) and pH values (pH 3–pH 12) and a narrow host range against *S. epidermidis*. Potent antimicrobial and antibiofilm activities were observed when the phage was tested against a highly susceptible bacterial isolate. These encouraging results open the door to new therapeutic opportunities in the fight against resilient biofilm-associated infections caused by *S. epidermidis*.

## 1. Introduction

*Staphylococcus epidermidis*, part of the natural microflora of human skin and mucosa, is an important opportunistic pathogen and the most frequent source of biofilm infections associated with the use of indwelling medical devices (e.g., catheter systems, prosthetic joints, and a range of other polymer and metal implants) [1,2]. These infections are usually caused by a breach in the skin barrier resulting from the insertion of the medical device, thereby allowing *S. epidermidis* to penetrate the host tissues [3].

Biofilms formation on foreign bodies protects bacteria from antibiotics and the immune system and is key to disease progression [4]. In addition, the global spread of multidrug-resistant lineages [5] may pose a major challenge to the treatment of *S. epidermidis* biofilm-associated infections, calling for an urgent need to identify alternative bacterial targets for the development of novel anti-infective strategies.

One promising approach is the use of bacteria-specific viruses, known as bacteriophages or phages, to control infections by pathogenic bacteria. Phages have been used since their discovery to treat bacterial infections in humans [6] and are currently suggested as possible alternatives or adjuncts to antibiotics for the treatment of bacterial diseases to minimize pathogen loads on medical devices [7].

The nature of phages advocates for their use in the treatment of infections caused by antibiotic-resistant, biofilm-producing bacteria. A phage’s mechanism of action and targeted receptors on the bacterial cell envelope are distinct from antibiotics, suggesting a low risk of phage-antibiotic cross resistance in bacteria [8]. Additionally, the high phage specificity, some of which can be up to the subspecies level, is often considered an advantage in phage therapy, whereby the potential for disruption of the microbiome is limited, as off-target commensals should not be infected [9]. Furthermore, some phages contain enzymes with polysaccharide depolymerization activity that can degrade bacterial biofilms [10].

Most of the staphylococcal phages isolated to date have been characterized in *Staphylococcus aureus* strains constituting a gap of knowledge on phages active against *S. epidermidis*. Moreover, there is only a limited number of studies investigating phage activity against *S. epidermidis* biofilms [11,12,13,14]. In this study, we isolated and characterized a novel bacteriophage specific to *S. epidermidis* and provided key data on different infectivity parameters of the phage as well as in vitro assessment of antimicrobial efficacy.

## 2. Materials and Methods

### 2.1. Bacterial Strains and Bacteriophage

This study comprised 29 clinical *Staphylococcus epidermidis* isolates obtained from patients diagnosed with implant-associated infections between 2015 and 2020. Bacteria were stored at −80 °C using a cryovial bead preservation system (Microbank; Pro-Lab Diagnostics, Canada). The novel bacteriophage CUB-EPI_14 targeting *S. epidermidis* was isolated from wastewater treatment plant sewage and further characterized.

Phage isolation was performed using the enrichment method as previously described [15]. To obtain a single pure phage, three consecutive single-plaque isolation cycles were performed on the bacterial host strain (SE14). Subsequently, a high titer phage solution was produced by propagation in liquid culture. Briefly, the *S. epidermidis* bacterial host was grown in tryptic soy broth (TSB) (US Biological, Eching, Germany) overnight at 37 °C. Using a sterile glass Pasteur pipette, a single phage plaque was picked, resuspended into a microcentrifuge tube containing 1 mL of filter-sterilized saline magnesium buffer (SM-buffer; 10 mM Tris-HCl, pH 7.8, 1 mM MgSO_4_), and incubated for 1 h at 4 °C. A volume of 0.2 mL of the overnight culture was inoculated into 20 mL sterile TSB and incubated with agitation at 37 °C till reaching an OD_600_ of 0.4 prior to the addition of 0.1 mL SM-buffer containing the resuspended phage plaque. The culture was then incubated at 37 °C with agitation for ~5 h or until the lysate was clear. Finally, the phage lysate was centrifuged at 4000× *g* for 20 min, and the supernatant was carefully collected and filter-sterilized using a 0.22 μm filter.

The filtered phage lysate was precipitated in the presence of 8% *w*/*v* polyethylene glycol (PEG-8000; PanReac AppliChem, Darmstadt, Germany) at 4 °C overnight, then centrifuged at 13,000× *g*, 4 °C for 40 min by using a fixed-angle rotor Eppendorf 5810 R centrifuge (Eppendorf, Hamburg, Germany). Thereafter, the pellet was resuspended in SM-buffer overnight, filtered, and stored at 4 °C until further use. Titration was performed on the host strain to determine the corresponding titer of the phage stock.

### 2.2. Morphological Analysis by Transmission Electron Microscopy

The morphology of CUB-EPI_14 was detected by transmission electron microscopy (TEM) using negative staining. Ten randomly selected viral particles were evaluated for size calculation. An aliquot of 15 µL of the high-titer phage particle preparation was dropped onto parafilm before the transferal onto a Ni-mesh grid (G2430N; Plano GmbH, Wetzlar, Germany), which has been carbon-coated and glow discharged (Leica MED 020, Leica Microsystems, Wetzlar, Germany), and allowed to adsorb for 10–15 min at room temperature. The grids were washed three times with Aquadest and subsequently treated with 1% aqueous uranyl acetate (SERVA Electrophoresis GmbH, Heidelberg, Germany) for 20 s for negative staining, followed by the removal of excess staining with filter paper. Grids were air-dried and then imaged by TEM using a Zeiss EM 906 microscope (Carl Zeiss Microscopy Deutschland; Oberkochen, Germany) at a voltage of 80 kV. Phage size measurements were calculated using the image processing software ImageJ.JS 1.53 m (https://imagej.nih.gov/ij/, accessed on 2 May 2022) [16].

### 2.3. Genome Extraction, Sequencing, and Annotation

To isolate the phage genome, 1 mL of phage stock was treated with 10 µg DNaseI and 50 µg RNaseA (Roche Diagnostics; Mannheim, Germany) in the presence of MgCl_2_ to degrade DNA that is not protected by a viral capsid, followed by 50 µg/mL of proteinase K (Thermo Scientific, Waltham, MA, USA), 20 mM EDTA and 0.5% SDS treatment to inactivate the DNaseI/RNaseA and disrupt the viral capsid. Subsequently, extraction by phenol-chloroform was performed to remove debris. The nucleic acid pellet was precipitated (14,000× *g*, 20 min) in the presence of absolute alcohol and washed with 70% alcohol before being suspended in deionized distilled water. Nanodrop measurements (Peqlab; Erlangen, Germany) were done to determine concentration and purity (260/230 ratio).

Sequencing was performed on an Illumina (San Diego, CA, USA) MiniSeq instrument. The Nextera Flex DNA library kit (Illumina) was used for library preparation. After assembly of the raw sequencing data using Unicycler [17] on the PATRIC v3.6.6 server [18], the most related phages were identified with BLASTn [19] and Viptree v1.9 [20]. VIRIDIC [21] was used for taxonomic classification. Next, annotation was performed with RASTtk [22] on the PATRIC server, followed by manual curation using BLASTp (and HMMER [23] or Phyre2 [24] in some cases). A genome map was visualized with Easyfig [25]. The data were submitted to NCBI GenBank under accession number ON325435.

### 2.4. Host Range and Efficacy of Plating

The host range of this novel phage was evaluated against 29 *S. epidermidis* and 74 *Staphylococcus aureus* strains by soft agar overlay spot assay. A bacterial lawn was prepared by mixing 50 µL of an overnight grown bacterial culture with 3 mL soft agar and pouring over tryptic soy agar (TSA) and letting 10 min to dry. Tenfold phage dilutions (10^−1^ to 10^−9^) were then spotted (5 µL) on the bacterial lawn and let to dry. Single phage plaque identification was performed after overnight incubation at 37 °C. Bacterial strains were classified as susceptible to the phage when single phage plaques were visible within any of the dilutions.

The bacterial susceptibility to CUB-EPI_14 was evaluated in terms of the efficacy of plating (EOP) as previously described [26]. The EOP value was determined by dividing the plaque-forming units (PFU) on the tested clinical strains by the phage titer on the host bacterium (EOP = phage titer on test bacterium/phage titer on host bacterium). EOP values of >0.5 were regarded ‘high’ efficiency; 0.2–0.5 as ‘medium’ efficiency; 0.001–0.2 as ‘low’ efficiency; 0.0 was considered as not effective against the target strain [27].

### 2.5. Adsorption and One-Step Growth Curve of CUB-EPI_14

The adsorption and one-step growth curve assays were performed to determine the adsorption rate, latent period, and burst size of CUB-EPI_14 as previously described [28]. Briefly, three *S. epidermidis* strains susceptible to the phage (SE14 [host], SE16, and SE18) were grown in 10 mL TSB at 37 °C under shacking conditions (150 rpm/min) to reach an OD_600_ of 0.4 (equivalent to approx. 10^8^ CFU/mL). The CUB-EPI_14 was then mixed with the bacterial strains at MOI = 0.01 (10^6^ PFU/mL) and incubated at 37 °C. Aliquots of 1 mL were taken at 10 min intervals for up to 60 min and centrifuged at 4000× *g* for 1 min to pellet adsorbed phages. Serial dilutions of the supernatant (non-adsorbed phage) were titrated to determine the number of non-adsorbed phage particles at each incubation time. 

The one-step growth curve was used to compute the phage latent time and burst size. Exponentially growing bacteria were infected with CUB-EPI_14 (MOI of 0.01) and incubated for 10, 20, or 40 min at 37 °C to allow phage particles to adsorb to SE14, SE18, and SE16, respectively. Then, the culture was centrifuged at 7000× *g* for 10 min and the pellet was resuspended in 10 mL of TSB and kept at 37 °C. Aliquots were removed at 20 min intervals for up to 280 min and titrated against the respective bacterial strain. The latent period was defined as the time taken for the phage particles to replicate inside the bacterial cells (i.e., from adsorption to first cell burst). Burst size was defined as the number of phage particles released from a single infected bacterial cell and was calculated by dividing the number of phages formed during the rise period by the estimated number of infected cells present in the culture at the latent period time (assuming that infected cells do not multiply or lyse from infection till rise period), as described by Nabergoj D et al. [29].

### 2.6. Thermal and pH Stability

The thermostability of CUB-EPI_14 was evaluated by incubating 10^8^ PFU/mL phage solution at various temperatures (−20 °C, 4 °C, 25 °C, 37 °C, 50 °C, and 60 °C) for 1 h and 24 h. Similarly, 10^8^ PFU/mL of phage solution was incubated in phosphate-buffered saline adjusted to different pH, ranging from 1 to 13 at room temperature (25 °C), for 1 h and 24 h to determine their stability at different pH levels. The soft agar overlay technique was used to titer phage samples in the host bacterium.

### 2.7. Antimicrobial Susceptibility Test by Isothermal Microcalorimetry

Isothermal microcalorimetry was used to determine the antimicrobial activity of CUB-EPI_14 against the host bacterium (SE14) and two other susceptible strains (SE16 and SE18), as previously reported [30]. Glass ampoules containing 0.8 mL of fresh TSB were inoculated with 0.1 mL of 10^5^ CFU/mL bacteria and 0.1 mL of phage (at titers ranging from 10^4^ to 10^9^ PFU/mL) and introduced into the calorimeter where heat-flow production was monitored at 37 °C for 48 h. A growth control (GC) without phage and two negative controls containing TSB only (NC) or TSB and phage (PC) without bacteria were included in every test.

### 2.8. Biofilm Time-Killing Assay

To evaluate the antibiofilm activity of CUB-EPI_14, a time-killing assay was performed by exposing 24-hour-old biofilm to the phage at different incubation times. Bacterial biofilms were formed on sterile 1.5 mm sintered porous glass beads (ROBU, Hattert, Germany) by incubation in a sterile 24-well plate (Corning Inc., Corning, NY, USA) containing 1 mL TSB inoculated with 1:100 dilution of *S. epidermidis* from a one-time use glycerol stock and kept at 37 °C and 150 rpm for 24 h under humid conditions. Subsequently, glass beads were washed three times with sterile 0.9% saline to remove non-adherent planktonic cells before being transferred to microcentrifuge tubes containing 1 mL of fresh TSB inoculated with CUB-EPI_14 (10^8^ PFU/mL). Samples were incubated at 37 °C for 0 h, 2 h, 4 h, 6 h, 8 h, 10 h, 12 h, and 24 h. After the corresponding incubation time, biofilm-embedded cells were retrieved by sonicating the glass beads in an ultrasound bath at 40 kHz and 0.2 W/cm^2^ (BactoSonic, BANDELIN electronic, Berlin, Germany) for 10 min. Then, the bead was removed, and the sonicated fluid was centrifuged (1 min, 16,000× *g*, 4 °C). The supernatant was discarded, and the bacterial pellet was resuspended in 1 mL 0.9% saline. This step was repeated three times to wash out the remaining phage particles from the samples. Then, ten-fold serial dilutions were plated onto TSA, and after 18–24 h incubation at 37 °C, recovered biofilm cells were quantified by colony counting. Data were plotted as bacterial count (CFU/mL) over time (h) using GraphPad Prism 6 software (GraphPad Software, La Jolla, CA, USA). Results were statistically analyzed using a multiple paired Student’s *t*-test analysis integrated into GraphPad Prism 6; *p*-values < 0.005 were considered significant (*).

### 2.9. Results Visualization and Analysis

All experiments were carried out in biological triplicates. Data were expressed as mean ± standard deviation (SD). GraphPad Prism 6 software (GraphPad Software, La Jolla, CA, USA) was used to prepare all plots. For visualization of calorimetric outcomes, plots of heat-flow (µW) over time (h) were prepared using representative samples.

## 3. Results

### 3.1. Bacteriophage Characterization

The morphology of CUB-EPI_14 was examined by TEM. The bacteriophage’s virion consisted of an icosahedral head (59.4 ± 1.5 nm in diameter), a non-contractile tail (237.9 ± 5.6 nm in length and 10.7 ± 0.7 nm in width) with no visible tail fibers (Figure 1), corresponding to a siphovirus morphology.

The phage’s host range was determined on 29 *S. epidermidis* and 74 *S. aureus* strains by soft agar overlay spot assay of tenfold serial dilutions. Only two *S. epidermidis* strains (SE16 and SE18), other than the host bacterium (SE14), were susceptible to CUB-EPI_14 infection, with EOP ratios ranging from 0.2 to 2.55 (Table 1), indicating medium to high lytic activity. None of the tested *S. aureus* strains showed susceptibility to the phage.

Over 95% of CUB-EPI_14 adsorbed to SE14 host bacterium within 10 min (Figure 2A). The average burst size was calculated to be 3 PFU per cell, with a 90-minute latent period (Figure 2B). Roughly 80% of the phage adsorbed to SE16 (Figure 2C) within 50 min before a rise in phage titer could be observed. During the one-step growth curve, the latent period lasted 90 min, and the burst size was estimated to be 92 PFU per cell (Figure 2D). For SE18, similarly to SE14, within 20 min incubation, almost 100% of the phages had been adsorbed, while the latent period and burst size determined in SE18 were identical as in SE14 (Figure 2E,F).

The genome of CUB-EPI_14 was determined using Illumina sequencing, showing a dsDNA genome of 46,098 bp, with a GC content of 35.02%, compared to its host. A search against the BLASTn database showed two significant hits: (1) the uncultured phage clone 9S_3 (Genbank accession number MF417888; 94% query cover; 96% sequence identity) identified in a South African study of a skin metavirome and (2) ct5pN1 (BK030923; 89% query cover; 94.86% identity) identified in a human metagenome study [31]. The CUB-EPI_14 proteome clusters to some extent with *Staphylococcus* siphoviruses vB_SepS_SEP9 (NC_023582) [32] and 6ec (NC_024355) [33]. To further classify this phage taxonomically, the intergenomic distance between the different related bacteriophages was calculated and plotted (Appendix A), revealing that *Staphylococcus* phage CUB-EPI_14 can be defined as a novel phage species belonging to a yet unclassified phage genus together with the uncultured phages 9S_3 and ct5pN1.

To determine the phage genome ends, the terminase protein sequence of CUB-EPI_14 was compared to terminases with known packaging strategies as previously described [34]. This approach suggested a headful packaging strategy, which was confirmed using PhageTerm [35], that predicted a *pac* site. The terminase gene was arbitrarily chosen as the genome start. Structural annotation of the genome identified 63 coding sequences all on the sense strand and no predicted tRNAs could be identified. Thirty-three coding sequences could be assigned a putative function (Figure 3). No lysogeny-associated proteins were identified, although the Phage.AI algorithm [36] predicted a temperate lifestyle. Moreover, no known virulence proteins were encoded on the phage genome (screened with VirulenceFinder [37]), making it potentially suitable for phage therapy use, even though its transduction potential should be further investigated. 

### 3.2. Stability of CUB-EPI_14 at Different pH and Temperatures

Figure 4 depicts phage CUB-EPI_14 thermal and pH stability. After 1 h incubation at temperatures ranging from −20 °C to 50 °C, no significant reduction in phage titer was observed, whereas after 24 h incubation at 50 °C and 60 °C active phage particles could no longer be detected. The pH stability of CUB-EPI_14 was determined across a wide pH range from 1 to 13. The results indicated a stable phage titer at pH levels ranging from 3 to 12 up to 24 h. At pH levels 1, 2, and 13, the phage was completely inactivated already after 1 h incubation.

### 3.3. Antimicrobial Activity

The antimicrobial effect of CUB-EPI_14 at increasing titers against the host bacterium (SE14) and SE16 and SE18 was assessed by monitoring the heat production during 48 h of treated samples in comparison to the untreated growth control (Figure 5, upper row). Complete growth inhibition of SE14 and SE18 (absence of heat production) was observed at MOI 1000 (10^8^ PFU/mL) and 10,000 (10^9^ PFU/mL), respectively, whereas at lower MOIs a dose-dependent effect could be observed with remarkable delay or decrease in the heat-flow (µW) production at increasing titers compared to the growth control. At MOI 100 (10^7^ PFU/mL) or higher, total growth inhibition of SE16 (lack of heat production) was detected. At lower MOIs, a significant decrease in the initial heat-flow (µW) peak compared to the growth control could be observed before the heat production raised at around 16 h from the start of the monitoring period.

According to the time-killing experiment (Figure 5, lower row), the highest reduction in bacteria (lack of CFU detection indicating a complete biofilm eradication) was observed after 10 h exposure of SE16 biofilm to CUB-EPI_14, with no biofilm re-growth up to 24 h after phage addition. On the contrary, exposure of SE14 and SE18 biofilms to CUB-EPI_14 revealed a 1-log reduction in CFU counts compared to the untreated sample at all measured time-points.

## 4. Discussion

Most studies on staphylococcal phages focus on phages infecting *S. aureus*, and only a few include phages infecting other *Staphylococcus* species such as *S. epidermidis*, encouraging the search for and study of phages active against *S. epidermidis* that may provide new insights for possible therapeutic use. 

Genome sequencing of phage CUB-EPI_14 revealed high similarity (86.5–92.2%) with two uncultured viruses in the NCBI database, leading to its definition as a novel phage species belonging to a yet unclassified phage genus. Based on whole-genome sequencing, there are no counterindications against using this phage in phage therapy.

Host range analysis of CUB-EPI_14, including *S. epidermidis* and *S. aureus* strains, showed a narrow host spectrum, as only two *S. epidermidis* strains (SE16 and SE18) could be infected with medium to high EOPs, beyond the isolation host strain (SE14). However, due to the limited number of bacterial strains of unknown diversity tested in this study, no broad conclusions on the host range can be extrapolated. In fact, a recent study has shown wide host ranges as the dominant trait among 94 different staphylococcal phage isolates, from both myo- and siphoviral morphology, as well as a temperate and virulent lifestyle [38].

One-step growth curves revealed two very distinct outcomes depending on the *S. epidermidis* strain used as the host during the experiment. Although the latent period was the same (90 min) in all three tested bacterial strains, the burst size differed substantially between SE16 (92 PFU/cell) and SE14 or SE18 (3 PFU/cell). Other studies have reported *S. epidermidis* phages with burst sizes ranging from 4.3 PFU/cell to 49.3 PFU/cell [12,32,38,39,40], of which siphoviruses showed burst sizes between 6.39 PFU/cell and 49.3 PFU/cell. Our study reports a lower and higher burst size than those previously observed.

As skin commensals, *S. epidermidis* survive and grow at a low temperature (<37 °C) and pH (~4.5–6.4), while in an infection environment, it faces higher temperature and pH values [3]. Thus, data on phage stability under different pH and temperature conditions might guide their potential application in clinical settings. Thermostability tests showed that CUB-EPI_14 was stable between −20 °C and 50 °C within 1 h, with a complete inactivation observed at 50 °C and 60 °C after 24 h. The effects of pH on the phage stability showed stable activity following incubation at conditions between pH 3 and pH 12, whereas it lost activity after incubation at conditions below pH 3 and above pH 12. These results would indicate a stable application of CUB-EPI_14 on the skin or at the infection site upon skin barrier breach.

The capacity of the newly isolated phage to inhibit *S. epidermidis* bacterial growth as well as to remove pre-established biofilms was tested using isothermal microcalorimetry and CFU-determination after sonication, respectively, against three susceptible isolates. CUB-EPI_14 showed higher bacterial growth inhibition and biofilm killing against SE16 than against the other two strains tested. In fact, the phage was able to eliminate (no detection of viable cells) the biofilm of SE16 after 10 h co-incubation. A possible reason for this outcome might be related to the higher burst size observed on SE16 that may lead to a rapid increase in the number of phage particles able to infect and kill bacterial cells within the biofilm, affecting the biofilm’s integrity, ultimately causing its complete degradation. Furthermore, analysis of the genome sequence with HHPred revealed a hit to a tail-tip associated hydrolase/lysin (Probability 99.84%, e-value 6.7 × 10^−20^), which next to helping the phage to inject its DNA into the host cell at the start infection, Gp17 potentially also degrades polysaccharides in the biofilm layer. Nevertheless, phage–biofilm interactions are complex, and the elimination of bacterial biofilms might depend on a variety of parameters such as susceptibility of the bacterial strain to the phage, initial phage titer, strength/structure of the biofilm, and phage entrapment and/or inactivation, among others [14,41,42].

Overall, the isolation and characterization of phages for the treatment of infections caused by the opportunistic *S. epidermidis* opens the door to new therapeutic opportunities, as biofilm-associated nosocomial infections are increasing in prevalence. Our study on the characterization of the new CUB_EPI-14 phage brings new insights for potential clinical applications, especially against resilient biofilm-associated infections. Future studies may address the effectiveness of the combined activity of this new phage with antibiotics against biofilms, as it has been shown that phages can act as adjuvants to antibiotic treatment, achieving better results compared to the action of each antimicrobial separately [43].

## Figures and Tables

**Figure 1 viruses-14-01340-f001:**
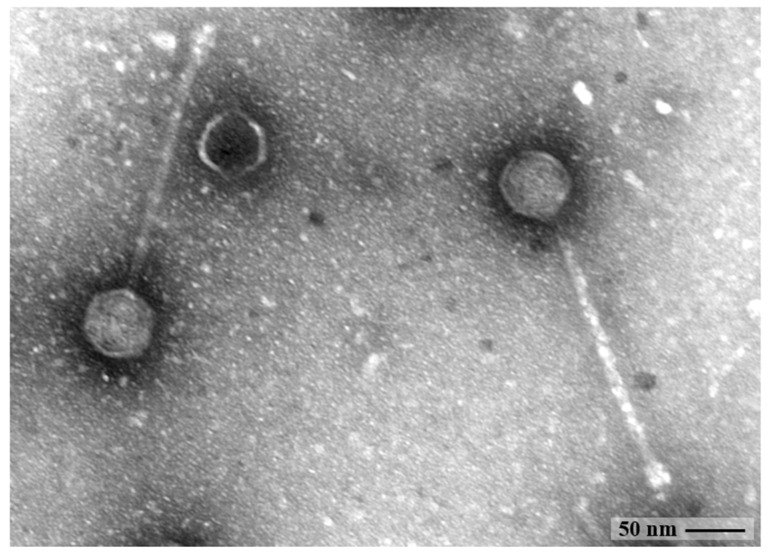
TEM image of the phage CUB-EPI_14 virions. Phage head (59 nm diameter) and tail (238 nm length and 10.7 nm width) measurements were determined with the image processing software ImageJ.JS [16].

**Figure 2 viruses-14-01340-f002:**
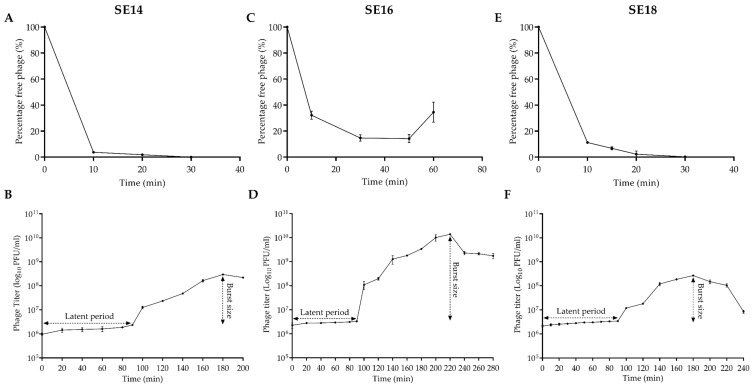
Adsorption (**upper row**) and one-step growth curves (**lower row**) of CUB-EPI_14 assessed on *S. epidermidis* strains SE14 (**A**,**B**), SE16 (**C**,**D**), and SE18 (**E**,**F**). Data are expressed as mean ± SD.

**Figure 3 viruses-14-01340-f003:**
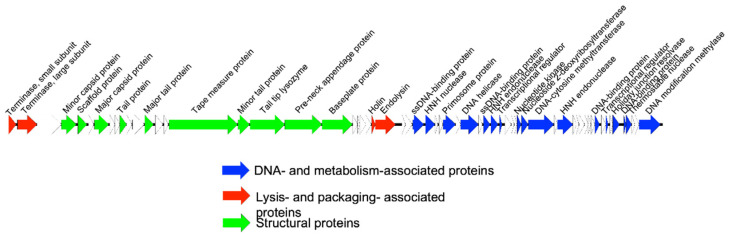
CUB-EPI_14 genome map. Each arrow represents a coding sequence. In red, genes encoding packaging and lysis-associated proteins are displayed; in green, structural proteins; and in blue, DNA- and metabolism-associated proteins (adapted from EasyFig).

**Figure 4 viruses-14-01340-f004:**
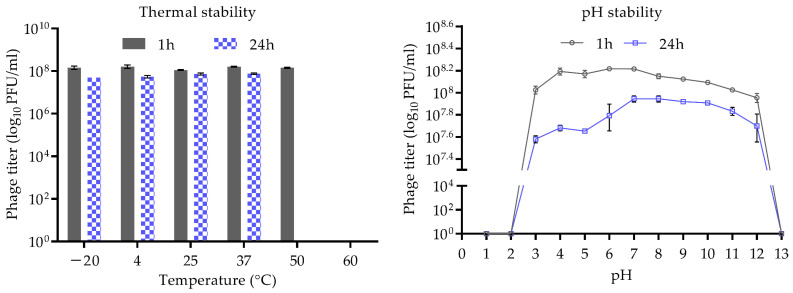
Thermal and pH stability test of CUB-EPI_14. Temperature experiments were performed for 1 h and 24 h at pH 7. pH experiments were performed for 1 h and 24 h at room temperature (25 °C). Error bars represent SD.

**Figure 5 viruses-14-01340-f005:**
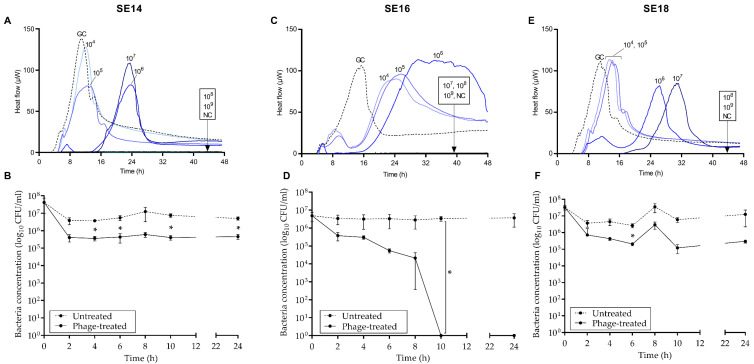
Microcalorimetric heat-flow curves (µW over time) of planktonic (10^5^ CFU/mL) SE14 (**A**), SE16 (**C**), and SE18 (**E**) co-incubated with phage CUB-EPI_14 at different titers (10^4^ to 10^9^ PFU/mL). GC, growth control; NC, negative control. Data of a representative experiment are reported. Time-killing curve of SE14 (**B**), SE16 (**D**), and SE18 (**F**) biofilms treated with CUB-EPI_14 (10^8^ PFU/mL) and untreated monitored at 2 h intervals for the first 10 h and after 24 h. Data are expressed as mean ± SD. Multiple paired Student’s *t*-test was performed; *p*-values < 0.005 were considered significant (*).

**Table 1 viruses-14-01340-t001:** Efficiency of plating (EOP) of CUB-EPI_14 phage dilutions against target bacteria. EOP values were determined using SE14 as reference strain. Values are expressed as average ± SD.

Strain	EOP	Rank
SE16	2.55 ± 1.6	high
SE18	0.2 ± 0.0001	medium/low

EOP values of >0.5 ranked as ‘high’ efficiency; 0.2–0.5 as ‘medium’ efficiency; 0.001–0.2 as ‘low’ efficiency; 0.0 was considered as not effective against the target strain.

## Data Availability

Bacteriophage’s whole genome sequence data presented in this study are openly available in the NCBI GenBank under accession number ON325435.

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
