# Peer review of "Novel Bacteriophage Specific against Staphylococcus epidermidis and with Antibiofilm Activity"

_viruses, 2022, doi:10.3390/v14061340_

Round 1

Reviewer 1 Report

General comments

This manuscript describes the characterization of a newly isolated Staphylococcus epidermis-specific bacteriophage. Based on the sequencing data obtained from this study, it is identified as a novel Staphylococcus phage closely clustering to Staphylococcus siphoviruses but belonging to a hitherto undescribed phage genus. The authors demonstrate the phage’s capacity to withstand a range of different temperatures and pH values and describe its narrow host range, able only to infect two of their 29 S. epidermis strains and none of their 74 S. aureus strains from their collection. Furthermore, they describe this phage’s ability to disrupt biofilm formation that often occurs on medical devices. Therefore, this novel study indicates that this phage could potentially be used, alone or in combination, to prevent infections caused by some S. epidermis strains related to indwelling medical devices. The study is straightforward, the manuscript is well written, and the data is clearly presented, requiring only minor changes in the text and the figures. Consequently, this article can be considered for publication in MDPI-Viruses following minor corrections.

Specific comments

-          Line 18: the text would flow better if “genomically” were written fully.

-          Line 21: replace “on the phage genome” with “in the phage genome”.

-          Line 44: replace “since they discovery” with “since their discovery”.

-    Line 68: can the authors clarify if the wastewater treatment plant was neighbouring CUB? And can the authors define how they developed the nomenclature of this phage? Obviously, CUB stands for Charité-Universitatsmedizin Berlin- and EPI must be for epidermidis. But is 14 representing the 14th phage isolated by this group? Or is it the 14th S. epidermidis phage in their collection?

-          Line 71: add the name of the host strain (SE14).

-          Line 126: change classify with classified in “bacterial strains were classified”.

-          Line 138: change sacking with shacking in “under shacking conditions”.

-          Lines 152-153: add “by” in “was calculated by dividing the number”.

-          Line 179: correct “under humid conditions”.

-          Lines 181-182: the B in TSB already stands for broth. So, you can remove broth at the beginning of line 182.

-          Line 186: change sonication with sonicated.

-          Lines 208: I would add: “Only two…” 

-          Line 217: change absorbed with adsorbed.

-          Lines 217-224: I would cite the figures in the order they appear in the text to make it easier to read and follow. Therefore, Figure 2A should come first, then Figure 2B, etc. So, the order of the labelling should change in Figure 2. The panels can be kept in the same way. However, Figure 2D can be labelled as Figure 2B, even though it is under Figure 2A. So, the top panel would have A, C, and E, and the lower panel would have B, D, and F.

-          Page 6, Figure 2 legend: correct raw for row as raw has a totally different meaning. Also, reorder the figures as described above. It would make it easier for the reader to add a label about the bacterial strain above each group of figures. So, adding a SE14 label above Figures 2A and 2B, SE16 above Figures 2C and 2D, and a SE18 title above Figures 2E and 2F.

-          Line 275: add “, respectively” after “10,000 (109 PFU/ml)”.

-          Page 8, Figure 5 legend: same comment as for the comment for the Figure 2 legend (and corresponding text) regarding the figure order.

-          Discussion: the authors mention in the introduction that some phages contain enzymes with polysaccharide depolymerization activity that can degrade bacterial biofilms. Since this manuscript is about a novel phage for which a genomic sequence has been generated and which can disrupt biofilm formation, can the authors discuss if they have found any genes encoding for such enzymes in phage CUB_EPI_14? And if not, what other mechanisms in their phage could be involved in biofilm destruction?

-          Figures 2 and 5: please increase the font size of both the x and y-axis to ease the axis’ legends’ reading.

-          Figures 2, 4 and 5: add statistical analyses between groups and timepoints, and a corresponding description in the materials and methods section of the statistical analyses used. I’m not a statistician, but perhaps a one-way or two-way ANOVA or student T-test between timepoints could confirm statistical significance (it’s better to check with a statistician about which test to use)?

Reviewer 2 Report

Pirlar et al report on the discovery and characterization of a new S. epidermidis phage and demonstrate its potential for use as a therapeutic. The phage has siphovirus morphology, yet lacks an identifiable integrase, making it potentially suitable for therapeutic applications. The authors demonstrate that the phage is stable over a range of temperatures and pH conditions, and shows anti-biofilm activity against a narrow range of hosts. Overall, the manuscript is straightforward and the authors report new and interesting findings. However, I do have a few comments for the authors to consider:

  1. Figure 1: To report accurate dimensions, at least 10 particles with negatively stained head should be measured across three axes, then averaged. Please clarify how many particles measured, and STDEV.
  2. Table 1: A table with the full range of hosts should be shown. Even though the host range appears narrow, it is unclear how closely-related the tested strains are.
  3. Figure 3: The presence of an RNA polymerase is not convincing as it seems too small. Please re-evaluate that gene call.
  4. Figure 5: The use of isothermal microcalorimetry to measure antimicrobial activity of the phage is unusual. Why not just show growth curves +/- phage? Without a standard growth curve, it is difficult to assess the impact of phage on the culture in this assay.

Minor:

Line 44: “their” instead of “they”

Line 138: please clarify what is meant by “sacking”

Line 199: “icosahedral” instead of “eicosahedral”

Line 331: “pre-established” instead of “pre-stablished”

Graphs in Figures 2, 4, and 5 appear to have low resolution and axis labels are difficult to read. Consider improving the resolution on these.
